# The Effects of the Follicle-Stimulating Hormone on Human Follicular Fluid-Derived Stromal Cells

**DOI:** 10.3390/ijms24032450

**Published:** 2023-01-26

**Authors:** Giedrė Skliutė, Brigita Vaigauskaitė-Mažeikienė, Algirdas Kaupinis, Mindaugas Valius, Edita Kazėnaitė, Rūta Navakauskienė

**Affiliations:** 1Department of Molecular Cell Biology, Institute of Biochemistry, Life Sciences Center, Vilnius University, Saulėtekio av. 7, LT-10257 Vilnius, Lithuania; 2Centre of Obstetrics and Gynaecology of the Institute of Clinical Medicine, Faculty of Medicine, Vilnius University, Santariškių St., LT-08661 Vilnius, Lithuania; 3Proteomic Center, Institute of Biochemistry, Life Sciences Center, Vilnius University, Saulėtekio av. 7, LT-10257 Vilnius, Lithuania; 4Faculty of Medicine, Vilnius University Hospital Santaros Klinikos, Vilnius University, Santariškių St., LT-08661 Vilnius, Lithuania

**Keywords:** follicle-stimulating hormone, follicular fluid, stromal cells, infertility, assisted reproductive technology

## Abstract

The prevalence of infertility is getting higher over the years. The increasing age of first-time parents, although economically more desirable, can cause various biological problems from low natural conception rate to poor pregnancy outcomes. The growing demand for assisted reproductive technology procedures worldwide draws medical specialists’ and scientists’ attention to various elements which could lead to successful conception, such as follicular fluid (FF) and hormones. In this study, we analyzed the effects of exposure to follicle-stimulating hormone (FSH) on FF-derived stromal cells isolated from females admitted for treatment due to infertility, participating in assisted reproductive technologies procedures. We demonstrated that FF stromal cells are positive for mesenchymal stromal cell surface markers (CD90^+^, CD44^+^, CD166^+^) and showed that FSH has no impact on FF stromal cell morphology yet lowers proliferation rate. Using a real-time polymerase chain reaction method, we indicated that the expression of *PTGS2* is significantly downregulated in FF sediment cells of patients who did not conceive; furthermore, we showed that FSH can affect the expression of ovarian follicle development and FSH response-related genes differentially depending on the length of exposure and that levels of ovulatory cascade genes differ in conceived and not-conceived patients’ FF stromal cells. Using mass spectrometry analysis, we identified 97 proteins secreted by FF stromal cells. The identified proteins are related to stress response, positive regulation of apoptotic cell clearance and embryo implantation.

## 1. Introduction

Infertility, often called a disease of the XXI century, is diagnosed when the couple is attempting to achieve a pregnancy for at least 12 months without conception. It is estimated that infertility can affect up to 15% of reproductive-age couples [1], although some studies have revealed that the prevalence was higher in couples who delay cohabitation with a partner, have higher socio-economic status and their parental age was higher when conceiving first child [2]. According to demographic and reproductive health surveys, primary infertility (the pregnancy has never been achieved) has ranged from 0.6% to 3.4%, while the range of secondary infertility (at least one prior pregnancy has been achieved) was from 8.7% to 32.6% [3]. Therefore, assisted reproductive technology (ART) cycles are gradually increasing worldwide [4] leading to more than 2% of babies birth in some European and USA countries as a result of ART [5,6].

When performing the ART cycle, clinicians receive a lot of information about couples’ fertilization process, cleavage, quality of embryos, etc. Nowadays, the fertilization process and quality of oocyte have been studied to understand what influences a good quality embryo formation and successful pregnancy. Follicular fluid (FF) has been investigated as it is responsible for the growth and maturation of the oocyte and could possibly affect the embryo’s development [7]. FF fills the cavity of the follicle and is produced under the influence of granulosa cells’ activity [8]. It contains proteins, hormones, fatty acids, enzymes, inflammatory factors, reactive oxygen species and other molecules [9]. It not only represents the environment of the oocyte but could be a target for biochemical markers research which could be clinically used in ART.

The follicle-stimulating hormone (FSH) is a hormone secreted in adenohypophysis under the control of the gonadotropin-releasing hormone. It is not only one of the main hormones regulating sexual development of the individual but also regulates the reproduction cycle through life [10]. For its important role in follicle growth and formation, recombinant FSH has been successfully used in ART treatment and helps many couples worldwide to conceive. When performing oocyte pick-up, it became clear that not all follicles contain an oocyte; thus, studies have been conducted to identify key factors, which could be associated with the recovery of the good quality oocyte. FSH has been found to be important when evaluating the recovery of the oocyte as it is considered to act in the physiologic event of separation of the cumulus-oocyte complex from the follicle wall [11]. Beyond its important role in ovulation, FSH participates in preantral follicles formation [12] by blocking apoptosis. Due to its important function in reproductive health and ART, we investigated its effect on follicular fluid-derived stromal cells.

To explore the possible effects of FSH on FF-derived stromal cells, we examined cell proliferation capacity, cell surface markers (CD90, CD44, HLA-ABC, CD166, CD73, CD13, CD105, CD56, CD146, BMSC, CD9, SUSD2, CD388, CD43, CD45 and CD140b) expression, ovarian follicle development and FSH response related genes *STAR, INHBA, INHBB, HAS2, PTGS2, LHCGR, HSD3B2, CYP11A1, CYP19A1, AMH, FSHR* expression levels and secreted protein profile in FSH-treated FF-derived stromal cells in females undergoing in vitro fertilization procedures. Detected molecular variations in FF-derived cells of females with unexplained infertility could enrich the understanding of underlying reasons for unexplained infertility and broaden the knowledge of defining qualities of follicular fluid cells.

## 2. Results

### 2.1. Study Population, the Clinical Data

To analyze the genetic profile of follicular fluid sediment cells and FSH-related molecular changes in follicular fluid-derived stromal cells, 16 females involved in ART were selected and divided into two groups according to IVF results: a conceived group (eight patients) and a not-conceived group (eight patients). Clinical data of study population are represented in Table 1. The hormones were investigated during 2–3 day of cycle before starting IVF. The average patient age in the conceived group was 30.6 years and in the not-conceived group 34.4 years. The average duration of infertility was 5.5 years in both study groups. In the conceived group, one female had endometriosis (Stage I) and two females had BMI > 25, whilst in the not-conceived group, three females had endometriosis (Stages I and II) and one female was overweight. The levels of FSH were compatible in both groups 5.6 U/L in the conceived group and 6.0 U/L in not conceived, levels of LH and estradiol did not differ significantly either: 4.5 U/L and 146.6 pmol/L in conceived patients and 3.4 U/L and 132.0 pmol/L in not-conceived patients. However, the levels of AMH were significantly decreased in not-conceived patients 2.1 µg/L, compared to conceived patients 4.6 µg/L. Moreover, significantly lower average amounts of oocytes and zygotes were obtained from not-conceived patients (11.6 and 7.8) compared to conceived patients (16.8 and 11.9). Overall, the clinical data suggest that not-conceived patients have a lower capacity to conceive due to the inferior number of oocytes and zygotes and reduced levels of AMH, signifying that there might be underlying molecular variations partially contributing to infertility.

### 2.2. Characterization of Follicular Fluid-Derived Stromal Cells Treated with FSH

We investigated the morphology, proliferation and expression of cell surface markers of control cells and cells treated with 200 pg/ul FSH for 30 days. We observed that the FSH treatment had no effect on FF stromal cell morphology (Figure 1A); however, treatment with FSH noticeably reduced PDL of FF stromal cells (Figure 1B).

FF stromal cells were further examined for the expression of surface markers CD90, CD44, HLA-ABC, CD166, CD73, CD13, CD105, CD56, CD146, BMSC, CD9, SUSD2, CD388, CD43, CD45 and CD140b (Figure 2). FF stromal cells were mainly positive for mesenchymal stromal cell surface markers (CD90^+^, CD44^+^, CD166^+^) and HLA-ABC^+^, with no noticeable effect of FSH treatment. Expression of CD73 was overall lower in conceived patients vs. not-conceived patients, thus FSH treatment increased the percentage of CD73^+^ cells. Expression of CD13 and CD146 did not change significantly in either of the investigated groups; however, there was a vast increase in the expression of CD105, CD56 and BMSC in not-conceived patients’ FF stromal cells after FSH treatment. There was scarcely any or no expression of SUSD2, CD338, CD140b and hematopoietic cell markers CD34 and CD45 in FF stromal cells.

### 2.3. Analysis of the Protein Expression in FF Stromal Cells

The levels of mesenchymal stromal cell intermediate filament vimentin remained stable comapred to the control cells to FSH-treated cells in conceived patients. Levels of vimentin seemed slightly higher in FSH-treated stromal cells than the control cells in the unconceived patient group (Figure 3A). Cadherin, the calcium-dependent cell adhesion protein, was upregulated comparing control vs. FSH-treated FF stromal cells in both, conceived and not-conceived patient group (Figure 3B).

### 2.4. Assessment of Gene Expression in FF Sediment Cells and FSH-Treated FF Stromal Cells

To characterize genetic profiles in FF sediment cells and evaluate the effects of FSH on FF stromal cells we analyzed the expression of ovarian follicle development-related genes (Figure 4), the prostaglandin-endoperoxide synthase 2 (*PTGS2*), the hyaluronan synthase 2 (*HAS2*), the steroidogenic acute regulatory protein coding gene (*STAR*), hydroxy-delta-5-steroid dehydrogenase, 3 beta- and steroid delta-isomerase 2 (*HSD3B2*), cytochrome P450 family 11 and 19 subfamily A member 1 (*CYP11A1* and *CYP19A1*), follicle-stimulating hormone receptor (*FSHR*), inhibin subunits beta A and B (*INHBA* and *INHBB*), anti-mullerian hormone gene (*AMH*) and luteinizing hormone/choriogonadotropin receptor gene (*LHCGR*). We detected significant downregulation of *PTGS2* in not-conceived FF sediment cells compared to conceived patients’ FF cells (Figure 4). Moreover, *HAS2, STAR*, *HSD3B2* and *CYP19A1* tend to be downregulated in the not-conceived patients’ group vs. the conceived patients’ group. FF sediment cells from conceived patients had a higher expression of *FSHR*, *AMH, INHBA* and *INHBB* and slightly lower expression of *CYP11A1* and *LHCGR* than not-conceived patients’ FF sediment cells, though none of these changes were statistically significant.

The effects of FSH treatment on the genetics of FF stromal cells in conceived and not-conceived patients were evaluated after 5, 10, 20 and 30 days of exposure to the substance (Figure 5). Levels of *PTGS2* significantly increased in conceived patients’ FF stromal cells after 5 days of 200 pg/mL FSH treatment and remained stable in 30 days period, expression of *HAS2* in conceived patients’ FF stromal cells was the most prominent after 5 and 10 days of FSH treatment and decreased at day 20. On the other hand, in not-conceived patients’ FF stromal cells *PTGS2* was upregulated after 5 and 10 days of FSH treatment, whilst levels of *HAS2* dropped throughout the 30-day exposure period. Expression levels of *STAR, HDS3B2* and *CYP19A1* remained stable in conceived patients’ FF stromal cells, whilst in not-conceived patients’ FF stromal cells *STAR* and *CYP19A1* were downregulated and HSD3B2 was upregulated during 30 days of FSH treatment. The *FSHR* gene had an early response to FSH treatment and increased after 5 days of exposure in both conceived and not-conceived patients’ cells, after 20 days it was elevated only in conceived patients’ cells and after 30 days of treatment, no expression of *FSHR* was detected. Furthermore, the expression of *LHCGR* was detectable only in control FF stromal cells in the not-conceived patients’ group, although in conceived patients’ FF stromal cells, the expression of *LHCGR* remained slightly elevated after 5 and 10 days of exposure to 200 pg/mL FSH. The expression level of *INHBA* was higher in conceived patients’ FF stromal cells after 5 days of FSH treatment, on the other hand, *INHBB* was more pronounced in not-conceived patients’ FF stromal cells after FSH treatment. FSH had no effect on conceived patients’ FF stromal cells *AMH* gene expression, yet, in not-conceived patients’ FF stromal cells *AMH* expression was elevated after 20 and 30 days of exposure to FSH.

### 2.5. Changes in FF Stromal Cells Secreted Proteins during FSH Treatment

Using the liquid chromatographic-mass spectrometry technique, we studied the secreted proteins of control and FSH-treated FF stromal cells of conceived and not-conceived patients experiencing infertility. The 97 human cell-specific secreted proteins were detected in FF stromal cell samples (Appendix A). Firstly, we assessed the secreted proteins of conceived patients’ FF stromal cells in control cells and cells treated with 200 pg/mL FSH for 30 days and compared results to secreted proteins of not-conceived patients’ FF stromal cells in control cells and cells treated with 200 pg/mL FSH for 30 days (Figure 6). We determined that proteins H2AC12, CFI, APOB, AFP, CPN1, TGFBI and RBP4 were upregulated in conceived patients’ FF stromal cell secretome after FSH treatment whilst being downregulated in FF stromal cell secretome of not-conceived patients’ FF stromal cells after FSH treatment. Moreover, we discovered that proteins PGK1, COL5A1, FBLN1, SERPINE1, TLN1, HSP90AA1, ITIH2, GAPDH, ITIH4, POTEI, FLNA, LOXL2, STC1, VIM, UCHL1, ACTN1, HSP90AB1, PLOD1, MYH9 and COL6A3, related to vinculin binding, negative regulation of transforming growth factor beta production and MHC class II protein complex binding, were downregulated in conceived patients’ FF stromal cell secretome after FSH treatment and upregulated in FF stromal cell secretome of not-conceived patients after FSH treatment. Comparing not-conceived control vs. conceived control FF stromal cells secreted proteins we discovered that 77% of proteins were upregulated and 23% were downregulated. Proteins that were more abundant in not-conceived FF stromal cell secretome were related to stress response, membrane-to-membrane docking, positive regulation of early endosome to late endosome transport, positive regulation of apoptotic cell clearance and embryo implantation, whilst less abundant proteins were involved in gluconeogenesis, negative regulation of angiogenesis and post-translational protein phosphorylation. Comparing not-conceived FSH-treated vs. conceived FSH-treated FF stromal cells secreted proteins we noticed that 80% of proteins were upregulated and 20% were downregulated. Proteins that were more prominent in not-conceived FSH-treated FF stromal cell secretome were related to the positive regulation of protein processing in the phagocytic vesicle, telomerase holoenzyme complex assembly and negative regulation of substrate adhesion-dependent cell spreading, whereas less copious proteins were involved in response to the hormone, protein oxidation and serine-type endopeptidase inhibitor activity.

## 3. Discussion

Studies show that follicular fluid contains various cell populations, differing in morphology and surface markers. By morphology alone, three main populations can be distinguished: fibroblast-like, epithelial-like and neuron-like cells [13]. Our focus was fibroblast-like cells–FF stromal cell population. It is known that a rise in FSH concentration at the end of the luteal phase stimulates granulosa cell maturation and proliferation and an increase in the amounts of FSH and LH receptors [14]. Therefore, we investigated the effects of FSH treatment on FF stromal cells. FSH is shown to stimulate the proliferation of epithelial ovarian cancer cells [15], ovarian surface epithelial cells [16], Sertoli cells [17] and ovarian granulosa cells, as shown in animal studies [18,19,20]. However, there are some reports of FSH inhibiting cell proliferation, for example, cervical cancer cells [21]. Our study revealed suppression of proliferation of FF stromal cells in presence of FSH. We showed that FF stromal cells were CD90^+^, CD44^+^, CD166^+^ and CD73^+^, with no impact on FSH or fertility status. Similar results were obtained by other groups, they showed that FF stromal cells are positive for CD44^+^ and CD105^+^ and were capable of osteogenic and adipogenic differentiation [13,22]. Mesenchymal stromal cells are known to be positive for vimentin expression, and FF stromal cells are proven to be no exception [23]. We also showed that FSH might somewhat promote the expression of vimentin and cadherin in conceived and not-conceived patients’ FF stromal cells.

Follicular fluid is typically investigated as the medium full of nutrients to stimulate cell growth from other biological sources, for example, endometrium-derived cells [24] or to investigate its impact on other cell types, such as fallopian tube epithelial cells [25]. However, there are lack of studies investigating genetics and proteomics of FF-derived cells, especially FF stromal cells. We showed that there is a significant decrease in the expression of *PTGS2* in not-conceived patients’ FF sediment cells, and FSH treatment increased levels of *PTGS2* in conceived and not-conceived patients’ FF stromal cells after 5 and 10 days of treatment. Lutz and colleagues showed the reduced expression of *PTGS2* in cumulus cells of infertile women with endometriosis [26]. Thus, low levels of *PTGS2* could indicate infertility. In addition, we discovered, that *HAS2, STAR* and *CYP19A1* tended to be downregulated in not-conceived patients’ FF sediment cells. FSH treatment increased the expression of *HAS2* in conceived patients’ FF stromal cells and the expression of *HSD3B2* was generally higher in not-conceived patients’ FF stromal cells than in conceived patients’ FF stromal cells after FSH treatment. Higher levels of intracellular cAMP and expression of ovulatory genes *AREG*, *EREG, HAS2* and *TNFAIP6* are shown to be regulated by PGE2, contributing to cumulus expansion and follicle rupture [27]. In a study of women with PCOS, it was shown that elevated expression of *VCAN, HAS2, PTX3* and *GREM1* could be associated to good oocyte quality [28]. Nonetheless, *TWNK, XRCC4/9* and *STAR* genes, related to adrenal and gonadal steroidogenesis, have been associated with syndromic primary ovarian insufficiency [29] and substantial increase in the expression of *HSD3B2* and *ESR*1 genes in follicular eutopic endometrium from infertile females with endometriosis was predicted to have a negative outcome on estradiol pathways [30].

It is known that activin and inhibin modulate folliculogenesis via the regulation of FSH secretion. Richani et al. showed that FSH increased levels of *INHA* and *INHBB* but decreased levels of activin B in granulosa cells of women undergoing IVF [31]. Other studies demonstrated decreased expression of *FSHR* and *CYP11A1* and increased expression of *CYP19A1*, *STAR*, *HSD3B2* and *INHBA* in polycystic ovary granulosa cells compared to controls [32]. Partial or total inactivation of FSHR, caused by the abnormal structure of the receptor, can be found in various disorders, such as infertility, amenorrhea and premature ovarian failure [33]. We demonstrated a higher expression of *FSHR*, *INHBA* and *INHBB* genes in FF sediment cells and a lower expression of *AMH, CYP11A1* and *LHCGR* in FF from conceived patients than not-conceived patients’ FF sediment cells. *FSHR* gene expression had a time-dependent response to FSH treatment—increased after 5 days of exposure and after 30 days of treatment no expression of the *FSHR* gene was detected. The expression level of *INHBA* was higher in conceived patients’ FF stromal cells after 5 days of treatment, whilst *INHBB* was elevated in not-conceived patients’ FF stromal cells after FSH treatment. AMH contributes to the assortment of the dominant follicle via regulating the sensitivity to FSH of preantral and antral follicles and constraining their growth. De Conto and colleagues showed that levels of *AMH* do not correlate with endometriosis-related infertility since expression of *AMH* in cumulus cells from control females and females with infertility were equivalent [34]. *LHCGR* is mainly expressed on granulosa cells of the preovulatory follicle [35]. Inherited mutations of *LHCGR* are shown to be related to oligo-ovulation or empty follicle syndrome [36]. In our study, *AMH* expression was elevated after long exposure to FSH in not-conceived patients’ cells and FSH suppressed the expression of *LHCGR* in FF stromal cells in not-conceived patients’ group, though in conceived patients ’FF stromal cells, the expression of *LHCGR* remained elevated after exposure. Discovered gene expression profile changes in FF sediment and stromal cells could help to deepen understanding of the relevance of FSH on FF stromal cells and give insights into underlying reasons for the difference in the molecular background of infertility patients and conception outcomes.

There are many studies investigating proteomics of follicular fluid. Zamah and colleagues identified 742 follicular fluid proteins in healthy donors, 413 of which there newly discovered. Discovered proteins were of receptor signaling, insulin, growth factor families and matrix metalloprotease-related proteins [37]. FF proteomics are also prevalent in disease research. Regiani et al. found 40 proteins unique to endometriosis patients’ FF. The endometriosis group proteins were involved in sterol metabolism, coagulation processes and responses to reactive oxygen species [38]. Ambekar et al., analyzed FF proteomic profile from females who were diagnosed with PCOS. They discovered 186 differentially abundant proteins in PCOS patients’ FF. In PCOS patients’ FF, proteins involved in the growth of the follicle AREG, HSPG2, TNF, TNFAIP6, PLG and LYVE1 were deregulated [39]. Our analysis of the secreted proteins of FSH-treated FF stromal cells of conceived and not-conceived patients revealed that the secretion of seven proteins were increased in conceived patients’ FF stromal cells after FSH treatment and the secretion of 20 proteins were decreased in conceived patients’ FF stromal cells after FSH treatment. Differently regulated proteins were involved in vinculin binding, negative regulation of transforming growth factor beta production and MHC class II protein complex binding and might be prevalent in folliculogenesis corresponding to lower numbers of retrieved oocytes in not-conceived patients with infertility. In summary, our findings contribute to the deepening of knowledge about FF-derived cells functioning and broaden our understanding of the FSH effect on a cellular level and the importance of FF to oocyte itself.

## 4. Materials and Methods

### 4.1. Study Population and Data Collection

We have conducted an interventional prospective cohort study. The patients were enrolled from 1 June 2021 to 30 December 2021 at the Vilnius University Hospital San-taros Clinics Fertility Center. Females with several infertility diagnoses, who received ART, entered the study. The inclusion criteria were as follows: (1) the age of women at the time of enrollment was 25–35 years, (2) the infertility diagnosis was confirmed after laboratory and instrumental investigation, (3) the women were diagnosed with primary infertility and (4) informed consent of all the subjects was received. The exclusion criteria were as follows: (1) the COVID-19 infection was confirmed during the treatment, (2) oncological disease was confirmed for women during the last three years, (3) women who were addicted to alcohol or other substances and (4) uncontrolled endocrine or other medical conditions, such as prolactinemia or thyroid diseases. The characteristics of the study population are presented in Table 1.

**Table 1 ijms-24-02450-t001:** Clinical data of the study population.

Characteristic	Conceived	Not Conceived
Patient count (n)	8	8
Average Age (years)	30.6 ± 4.7	34.4 ± 2.7
Duration of infertility (years)	5.5 ± 3.8	5.5 ± 3.4
FSH U/L	5.9 ± 1.6	6.0 ± 1.7
AMH µg/L	4.6 ± 2.5	2.1 ± 1.5
LH U/L	4.5 ± 1.4	3.4 ± 0.7
Estradiol pmol/L	146.6 ± 86.1	132.0 ± 47.1
Average of oocytes retrieved (n)	16.8 ± 4.9	11.6 ± 4.2
Average of zygotes (n)	11.9 ± 3.2	7.8 ± 4.3
BMI > 25 (n)	2	1
Diagnosis (n)	Tubal factor infertility (2) Unexplained infertility (5) Male factor infertility (1)	Tubal factor infertility (2) Unexplained infertility (4) Male factor infertility (2)
Endometriosis (n)	Stage I (2)	Stage I (1) Stage II (1)

n, number of females with a trait; BMI, body mass index.

### 4.2. Collection of Follicular Fluid, Extraction of Sediment Cells and Cultivation, Proliferation and FSH Treatment of Follicular Fluid-Derived Stromal Cells

The follicular fluid-containing heterogenic population was collected at the time of oocyte aspiration without flushing (36 h after human chorionic gonadotropin (6500 IU) trigger administration) into sterile tubes. Follicular fluid samples used for analysis were macroscopically clear and not contaminated with blood. Once collected, follicular fluid was transferred into 50 mL tubes and centrifuged at 500× *g* for 10 min, the supernatant was removed and 20 mL of 1× red blood cell (RBC) lysis buffer (10×, 155 mM NH_4_Cl (Sigma-Aldrich, St. Louis, MO, USA), 12 mM NaHCO3 (Sigma-Aldrich, St. Louis, MO, USA), 0.1 mM EDTA (Sigma-Aldrich, St. Louis, MO, USA), pH 7.3) added to the pellet, incubated for 5 min at room temperature and centrifuged at 500× *g* for 5 min. The isolated heterogenic cell pellet was divided: half was mixed with DNA/RNA lysis buffer (Zymo research, Irvine, CA, USA) and the other half was washed in phosphate-buffered saline (PBS) (Gibco, Thermo Fisher Scientific, Waltham, MA, USA) twice and cells counted. Cells were suspended in growth medium DMEM/F12 medium (Gibco, Thermo Fisher Scientific, Waltham, MA, USA) with 10% FBS (Gibco, Thermo Fisher Scientific, Waltham, MA, USA) and 1% penicillin (100 U/mL)—streptomycin (100 µg/mL) solution (Gibco, Thermo Fisher Scientific, Waltham, MA, USA), seeded into plastic cell culture flasks and cultivated at 37 °C in a humidified 5% CO_2_ atmosphere. Following 2–4 days of culture, non-adherent cells were washed away with PBS (Gibco, Thermo Fisher Scientific, Waltham, MA, USA) and adherent fibroblast-like cells were cultivated. Adherent cell viability and proliferation were evaluated by the trypan blue test. The cumulative population doubling level (PDL) was estimated as follows: PDL(X) = 3.32(log(total viable cells harvested)/(total viable cells seeded)) + PDL(X-1 passage). Follicle-Stimulating Hormone (FSH) treatment was performed for 5, 10, 20 and 30 days supplementing the growth medium with 200 pg/mL of recombinant human FSH (ProSpec, Rehovot, Israel), the medium was refreshed every 3–4 days.

### 4.3. Evaluation of Cell Surface Markers via Flow Cytometry Analysis

For phenotypical characterization of follicular fluid stromal cells, 0.05 × 10^6^ cells per assay were collected by centrifugation at 500× *g* for 5 min. The cell pellet was washed twice in PBS with 1% bovine serum albumin (BSA) (Sigma-Aldrich, St. Louis, MO, USA). Then, cells were suspended in 50 μL of PBS containing 1% BSA and incubated with the antibodies against cell surface markers in the dark at 4 °C for 30 min. Antibody dilution 1:25. Antibodies are presented in Appendix A. After incubation samples were washed with PBS with 1% BSA, cells suspended in 200 μL PBS with 1% BSA and analyzed using Guava easyCyte 8HT flow cytometer (Millipore, Burlington, MA, USA) using GuavaSoft™ 3.3 software (Luminex Corporation, Austin, TX, USA).

### 4.4. Gene Expression Analysis via RT-qPCR

The total RNA from sediment cells was purified using Quick-DNA/RNA™ Miniprep Kit (Zymo research, Irvine, CA, USA) and from stromal cells using TRIzol reagent (Invitrogen, Carlsbad, CA, USA), cDNA was synthesized using LunaScript^®^ RT SuperMix Kit (New England Biolabs, Ipswich, MA, USA), and qPCR was performed using Luna^®^ Universal qPCR Master Mix (New England Biolabs, Ipswich, MA, USA) on the RotorGene 6000 system (Corbett Life Science, QIAGEN, Hilden, Germany). Primer sequences (Metabion international AG, Planegg/Steinkirchen, Germany) are presented in Table 2. mRNA levels were normalized to *RLP13A* expression. Relative gene expression was calculated using the ΔΔCt method.

### 4.5. Proteins Analysis by Fluorescent Microscopy

FF stromal cells were seeded on the coverslips and fixed for 15 min with PBS + 4% paraformaldehyde solution at room temperature. Later, samples were washed with PBS and permeabilized using 10% Triton X-100/PBS for 20 min, then, washed in PBS again. Cells were blocked with PBS+2% BSA for 30 min at 37 °C. To detect expression of cadherin and vimentin the coverslips were incubated with primary antibodies against vimentin and cadherin and secondary goat anti-rabbit IgG, Alexa Fluor-488 antibody (Thermo Fisher Scientific, Waltham, MA, USA) in a humid chamber for 1 h at 37 °C. Actin-phalloidin was labeled with Alexa Fluor-594 Phalloidin (Thermo Fisher Scientific, Waltham, MA, USA) for 30 min at room temperature in a humid chamber. After each incubation, coverslips were washed with PBS+1% BSA. Nuclei were stained with 300 nM DAPI (Invitrogen, Waltham, MA, USA). Coverslips were mounted using Dako Fluorescent Mounting Medium (Agilent Technologies, St. Clara, CA, USA). Samples were analyzed using Zeiss Axio Observer (Carl Zeiss AG, Oberkochen, Germany) fluorescent microscopy system with the 63 X objective with immersion oil and Zen BLUE 12.0 software (Carl Zeiss AG, Oberkochen, Germany).

### 4.6. LC-MS-Based Protein Identification

The follicular fluid-derived stromal cell culture medium was aspirated and centrifuged for 10 min at 2000× *g* to remove cells and debris. After centrifugation, the supernatant was collected and filtered through a 0.22 μm filter. After filtration, secreted proteins were concentrated using the ProteoMiner^TM^ Sequential Elution Large-Capacity kit (Bio-Rad Laboratories, Hercules, CA, USA) according to the manufacturer’s instructions. Protein concentration was measured using the Pierce Detergent Compatible Bradford Assay Kit (Thermo Fisher Scientific, Waltham, MA, USA) according to the manufacturer’s instructions. The secretome proteins were trypsinized according to the FASP protocol. Briefly, proteins were diluted in 8 M urea; following two washes with urea, proteins were alkylated with 50 mM iodoacetamide (GE Healthcare Life Sciences, MA, USA). Protein concentrators were washed twice with urea and twice with 50 mM NH_4_HCO_3_ and proteins were digested overnight with TPCK Trypsin 20,233 (Thermo Scientific, Vilnius, Lithuania). After overnight digestion, peptides were collected from the concentrators by centrifugation at 14,000× *g* for 10 min and additionally eluted using 20% CH_3_CN. The eluates were combined, acidified with 10% CF_3_COOH and lyophilized in a vacuum centrifuge. The lyophilized peptides were redissolved in 0.1% formic acid.

Liquid chromatographic (LC) analysis was performed in a Waters Acquity ultra-performance LC system (Waters Corporation, Wilmslow, UK). Peptide separation was performed on an ACQUITY UPLC HSS T3 250 mm analytical column. Data were acquired using Synapt G2 mass spectrometer (MS) and Masslynx 4.1 software (Waters Corporation, Wilmslow, UK) in positive ion mode using data-independent acquisition (UDMS^E^). Raw data were lock mass-corrected using the doubly charged ion of [Glu1]-fibrinopeptide B (m/z 785.8426; [M+2H]2+). Raw data files were processed and searched using ProteinLynx Global SERVER (PLGS) version 3.0.1 (Waters Corporation, Wilmslow, UK). Data were analyzed using trypsin as the cleavage protease; one missed cleavage was allowed, and fixed modification was set to carbamidomethylation of cysteines; variable modification was set to oxidation of methionine. Minimum identification criteria included 1 fragment ions per peptide, 3 fragment ions and one peptide per protein. The following parameters were used to generate peak lists: (i) low energy threshold was set to 150 counts, (ii) elevated energy threshold was set to 50 counts, (iii) intensity threshold was set to 750 counts. UniprotKB/SwissProt human databases (11 October 2022) were used. Protein quantification was calculated using the ISOQuant software. Results of mass spectrometry analysis were evaluated using the online tool STRING (String consortium, https://string-db.org, accessed on 9 November 2022).

### 4.7. Statistical Analysis

Data were expressed as mean ± standard deviation (S.D.). Mann–Whitney U test and Two-way ANOVA Bonferroni’s multiple comparisons test were used to calculate the significance of difference; significance was set at *p* ≤ 0.01 (**); *p* ≤ 0.001 (***).

## Figures and Tables

**Figure 1 ijms-24-02450-f001:**
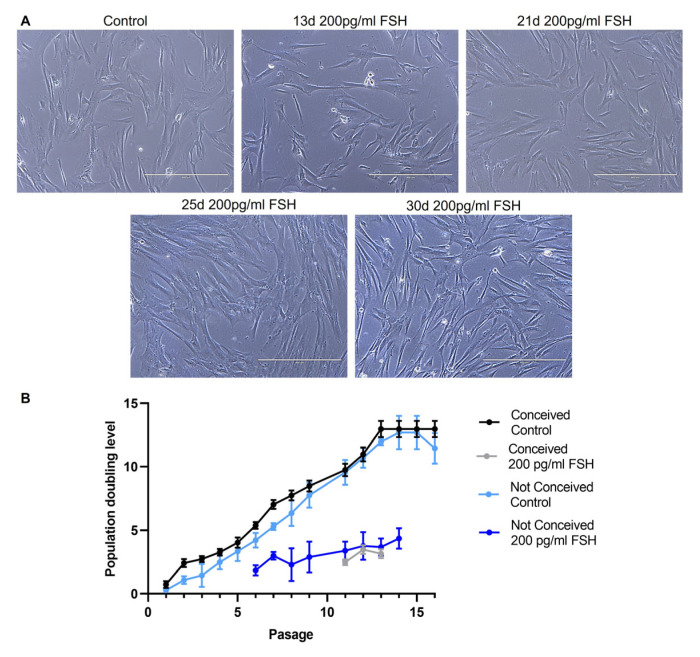
(**A**) FF stromal cell morphology. Control—untreated cells, 13d, 21d, 25d, 30d—cells after 13, 21, 25 and 30 days of culture in medium with 200 pg/mL FSH. Scale bar: 400 µm (10× objective).; (**B**) Population doubling level of FF stromal cells, passage represents number of times the cells have been subcultured.

**Figure 2 ijms-24-02450-f002:**
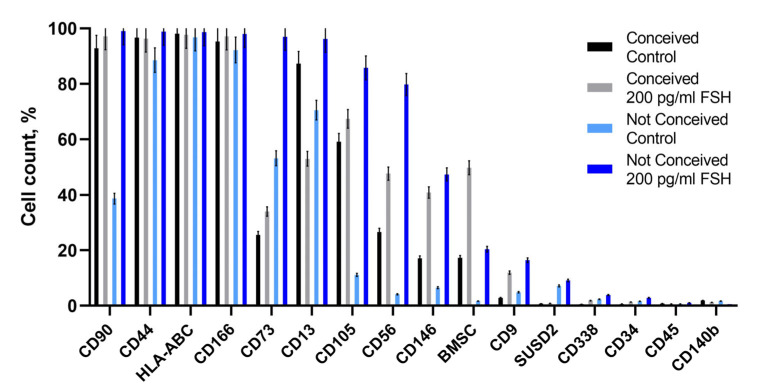
Analysis of cell surface markers of FF stromal cells. Control—untreated cells, FSH—cells after 30 days of culture in medium with 200 pg/mL FSH.

**Figure 3 ijms-24-02450-f003:**
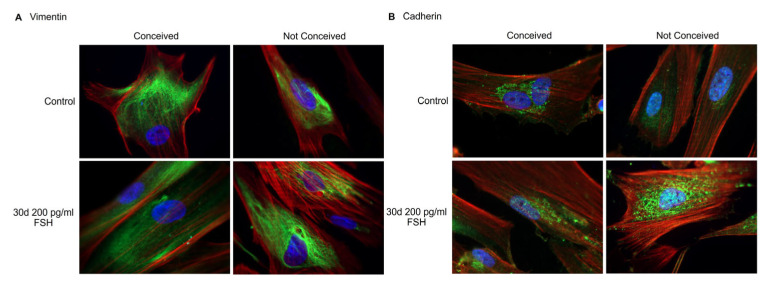
The levels and localization of vimentin protein (**A**) and cadherin protein (**B**) in control and 200 pg/mL FSH-treated FF stromal cells on day 30 of treatment. Nuclei are stained blue, actin-phalloidin is stained red, and vimentin and cadherin are stained green. Samples were analyzed using a Zeiss Axio Observer fluorescence microscope, with a 63× objective in immersion oil. Scale bar = 10 µm.

**Figure 4 ijms-24-02450-f004:**
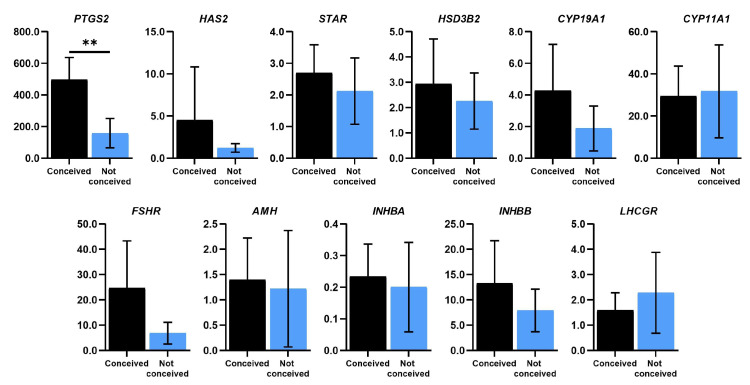
Real-time PCR analysis of genes related to ovarian follicle development and FSH response in cells of human follicular fluid sediment. Results are presented as mean ± standard deviation. In the conceived group, *n* = 6; in the not-conceived group, *n* = 6. ** *p* ≤ 0.01, based on the Mann–Whitney U test.

**Figure 5 ijms-24-02450-f005:**
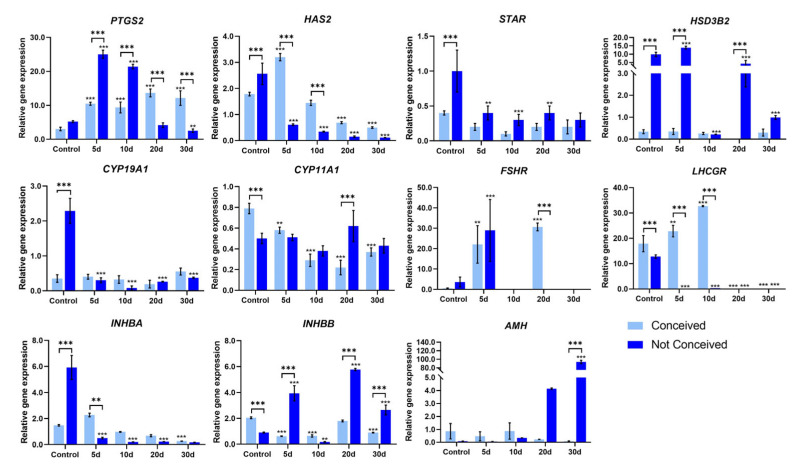
Real-time PCR analysis of genes related to FSH response in human follicular fluid stromal cells. Results are presented as mean ± standard deviation. In the conceived group, *n* = 3; in the not-conceived group, *n* = 3. Statistical significance calculated using the two-way ANOVA, *p* ≤ 0.01 (**); *p* ≤ 0.001 (***).

**Figure 6 ijms-24-02450-f006:**
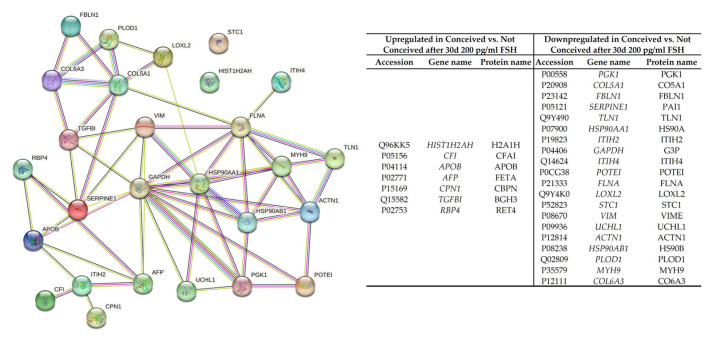
Differently secreted proteins in conceived and not-conceived patients’ FF stromal cells after 30 days of exposure to 200 pg/mL FSH. The interaction networks of secreted proteins and their involvement in different biological processes were identified using the STRING database, available online at https://string-db.org/ (accessed on 9 November 2022).

**Table 2 ijms-24-02450-t002:** List of primers sequence.

Name	Primer Sequence	Product Size, bp
*RPL13A*	F: GTTGATGCCTTCACAGCGTA	128
	R: AGATGGCGGAGGTGCAG	
*STAR*	F: GGCTCAGGAAGGACGAAGAACC	188
	R: ATCACAGCCTGTTGCCTCAGC	
*INHBA*	F: CCTCGGAGATCATCACG	238
	R: CCCTTTAAGCCCACTTCCTC	
*INHBB*	F: CCTGAAACTCCTGCCCTACG	104
	R: CCACCATGTTCCACCTGTCA	
*HAS2*	F: AGCCTTCAGAGCACTGGGACGA	81
	R: ACAGATGAGGCTGGGTCAAGCA	
*PTGS2*	F: TGAAACCCACTCCAAACACA	198
	R: AGGAGAGGTTAGAGAAGGCT	
*LHCGR*	F: TGGAGAAGATGCACAATGGA	122
	R: GGCAATTAGCCTCTGAATGG	
*HSD3B2*	F: TGCCTTGTGACAGGAGCAGG	238
	R: TACAGGCGGTGTGGATGACG	
*CYP11A1*	F: GTGATGACCTGTTCCGCTTTGC	155
	R: AAGGTTGAGCATGGGGACGC	
*CYP19A1*	F: GCTGGACACCTCTAACACGCT	289
	R: CAGGTCACCACGTTTCTCTGCT	
*AMH*	F: CGCCTGGTGGTCCTACAC	60
	R: GAACCTCAGCGAGGGTGTT	
*FSHR*	F: TGGGCTCAGGATGTCATCATCGGA	145
	R: TGGATGACTCGAAGCTTGGTGAGG	

## Data Availability

The data presented in this study are available on request from the corresponding author.

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
