# Peer review of "The Effects of the Follicle-Stimulating Hormone on Human Follicular Fluid-Derived Stromal Cells"

_ijms, 2023, doi:10.3390/ijms24032450_

Round 1
Reviewer 1 Report
In the present study, the authors isolated human follicular fluid (FF)-derived stromal cells. They mainly identified the marker of these cells and conducted liquid chromatographic-mass spectrometry to analyze the secreted-proteins. The data are of high quality and the paper is clearly written and well reasoned. However, this is also a straightforward paper with inadequate novel discoveries showing the importance of stromal cells in promoting infertility or assisted reproductive technology. Some results associated with mesenchymal surface markers and gene expression are predictable, and thus do not convey exciting formation. It is better to focus on one point and dig deeper.
I am very concerned about the chose of internal reference. As the GAPDH was selected throughout the manuscript, while in Result 2.5., we knew that GAPDH changed significantly between different groups. So it is unreasonable to set it as the internal reference to compare the relative levels of other molecules.
Minor
In figure 6, the table uses Protein Entry Name, but the left PPI image uses Gene Name, it makes the results hard to read.
Line 356, the inclusion criteria includes (3) the women who conceived after ART. Does it mean that both the conceived/not conceived group must have ever conceived before?
Reviewer 2 Report
The present study is a retrospective examination of the cells in the follicular fluid of the ovulatory follicle in women enrolled in ART because of infertility. Sixteen patients were assigned equally based on their pregnancy outcome to either conceived or not conceived groups. (Were 16 assigned ahead and happened to split into two equal-sized groups, or were eight selected who had conceived and eight selected who had not conceived? If the latter, how were patients chosen for each group?) Clinical data are descriptors of the cohorts, with characteristics previously noted by others for infertility patients. Table 1 does fit better with the description of the methods and not the results; no treatments were applied to the patients for hypothesis testing: delete p values. Were the total oocytes retrieved equivalent to 2.1 and 1.4 per patient? Were more follicles analyzed in the conceived group than in the not conceived?
A significant problem with the manuscript is the statistical analysis model for analyzing the data. The PIs have a 2x2 factorial experimental design with the factors being conceived and FSH treatment. However, you used a Mann-Whitney U test, which assumes only one variable is examined and does not test for interactions of two variables (conceived and FSH treatment) on responses. This does not fit the scope of the experiment, which was to determine if adding FSH to the incubation medium would differentially affect the responses in conceived versus not conceived patients. In the results, you imply interactions in responses to FSH or no FSH in conceived versus not conceived. Further, the authors should have stated why the entire group was only sampled for some analyses. Although eight patients were assigned to the groups, only six were used in each group for the real-time PCR analyses of genes related to follicular development in follicular fluid sediment cells (Figure 4) and three in each group for the real-time PCR analyses of follicular stromal cells (Figure 5). Only PTSG2 was significantly different between not conceived and conceived, and no genes were significantly different in Figure 5. Were the number of samples sufficient to describe characteristics of cells for conception versus no conception? The authors should have stated why the entire group was not sampled. The figure 1 legend does not agree with the x-axis label in the figure: are passages aligned to days in culture? Why is FSH treatment not begun at 1 and run through 16?
The manuscript should be improved by deleting colloquialisms such as “nowadays,” “all sorts of,” and “than ever before.” The manuscript should be edited to reduce wordiness, such as “in our research,” “in this study,” and “to begin with.” Lines 230-260 should be eliminated because they are not germane to the topic for discussion. Also, line 33 is misleading (speculative) because you did not examine the function of the cells or study the oocyte. The PIs assigned patients after knowing the outcome of ART, which precludes a prospective approach to pregnancy success.
Round 2
Reviewer 1 Report
Thank you for the authors and the current manuscript has been greatly improved.
My questions have been addressed.
Reviewer 2 Report
The investigators have responded well to the specific comments.